# The Efficacy of Positron Emission Tomography/Computed Tomography Scan (PET CT Scan) in the Diagnosis of Local Recurrence and Metastases in Surgical Patients with Medullary Thyroid Carcinoma: A Systematic Review and Meta-Analysis of the Last 5 Years (2020–2024)

**DOI:** 10.3390/cancers16244236

**Published:** 2024-12-19

**Authors:** Konstantinos Papadopoulos, Ioannis Iakovou, Stylianos Mantalovas, Christoforos S. Kosmidis, Stiliani Laskou, Vasileios Alexandros Karakousis, Christina Sevva, Marios Dagher, Panagiota Roulia, Ismini Kountouri, Isaak Kesisoglou, Konstantinos Sapalidis

**Affiliations:** 13rd Surgical Department, University General Hospital of Thessaloniki “AHEPA”, School of Medicine, Faculty of Health Sciences, Aristotle University of Thessaloniki, 1st St. Kiriakidi Street, 54621 Thessaloniki, Greece; steliosmantalobas@yahoo.gr (S.M.); kosmidisc@auth.gr (C.S.K.); stelaskou@gmail.com (S.L.); alexanderkarakousis@gmail.com (V.A.K.); christina.sevva@gmail.com (C.S.); mariosdag@gmail.com (M.D.); i.kountouri531@gmail.com (I.K.); ikesis@auth.gr (I.K.); sapalidis@auth.gr (K.S.); 22nd Academic Nuclear Medicine Department, Academic General Hospital of Thessaloniki “AHEPA”, Aristotle University of Thessaloniki, 54636 Thessaloniki, Greece; iiakovou@auth.gr

**Keywords:** medullary thyroid carcinoma, PET/CT scan, ROC curve

## Abstract

The management of medullary thyroid carcinoma (MTC) poses substantial challenges due to its high propensity for recurrence and metastasis, even after surgical intervention. Early detection of recurrence is crucial, yet traditional monitoring strategies, which are predominantly reliant on serum calcitonin levels and morphological imaging, exhibit notable limitations. In this landscape, the role of advanced imaging modalities, particularly PET/CT scanning, has become increasingly pivotal for accurately detecting and localizing recurrent or metastatic MTC, given their superior diagnostic precision. Results indicate that the sensitivity of PET/CT scanning, irrespective of methodology, stands at 82.8% with a 95% confidence interval of 75.3% to 88.3%, while the specificity is 91.5% with a 95% confidence interval of 81.2% to 96.4%. Moreover, a comparative analysis of the Gallium-68 DOTATATE (Ga68-DOTATATE) and Fluorodeoxyglucose (18-FDG) methods showed minor differences, with overlapping confidence intervals throughout. Statistical testing further supported this observation, with *p*-values of 0.619, 0.868, and 0.859 for sensitivity, specificity, and overall diagnostic accuracy, respectively.

## 1. Introduction

Medullary thyroid carcinoma (MTC) is a tumor that arises from the parafollicular cells of the telobranchial bodies (C cells), which secrete calcitonin and are of neuroendocrine origin, rather than from thyroid follicular cells. It is located predominantly at the upper poles and corresponds to a thyroid tumor due to its anatomical location [1]. It was first described by Hazard in 1959 and accounts for 5–10% of thyroid carcinomas, although new research has shown that it accounts for 1–2% and is the third most common type after papillary and follicular carcinomas [2]. It is distinguished in two forms [3]. In 75% of cases, the tumor is sporadic, with peak occurrence at 40–60 years of age, while in 25% it is hereditary and associated with the most common form of multiple endocrine neoplasia (MEN2A), which is MEN2B.As it is familial, itis now considered a variant of MEN2A, and it is transmitted in an autosomal dominant fashion [4].

MTC is a particularly aggressive carcinoma because while well-differentiated stage I-III thyroid carcinoma has a 10-year survival rate of 98.5–99%, this decreases to about 70% in medullary carcinoma. In addition, it presents with metastatic cervical adenopathy in 50% of cases at the time of diagnosis and with hematogenous distant metastases, such as in the lungs and bones, in 5–10% of cases [5]. Usually, sporadic carcinomas occur without multifocality as single cancerous foci in single lobes, in contrast to the hereditary type, which occurs with multifocality and is detected in younger age groups. Regarding MEN2A, it coexists in 50% of cases with pheochromocytoma and in about 20–25% of cases with primary hyperparathyroidism, as well as rare cases of cutaneous lichen amyloidosis and Hirschsprung’s disease. However, MEN2B, which is quite rarely detected, is accompanied by pheochromocytoma, and in a large percentage of patients it appears with megacolon [6]. In addition, among their differences, the former (MEN2A) occurs with higher frequency (95%) occurs in the absence of Marfan-like features; and is less aggressive, with lower mortality rates than MEN2B. For this reason, these pathological entities must be investigated before a surgical intervention in order to exclude any possible risk to the patient’s life, as in the case of pheochromocytoma, where the patient must first undergo adrenalectomy followed by total thyroidectomy, whereas if primary hyperparathyroidism is present, the surgeon may, during the total thyroidectomy, proceed to a subtotal parathyroidectomy, leaving a piece of the gland in situ a total parathyroidectomy with auto-transplantation of small slits of parathyroid tissue in a heterotopic position, such as the sternocleidomastoid; or resection of only the localized parathyroid adenoma, with intraoperative monitoring of parathyroid hormone (PTH) levels to document complete removal of the hyperfunctioning parathyroid tissue [7].

After surgical treatment of medullary thyroid carcinoma, local recurrence and metastatic disease are common conditions. Due to the increased incidence of local recurrence and the difficulty of reoperation, emphasis has been placed on postoperative screening of all these patients [8]. A cornerstone of their treatment is controlling calcitonin. Calcitonin has the most important prognostic role during follow-up [1,7,8,9]. Specifically, the positive predictive value of this method reaches 100% [10]. However, it is not possible to differentially diagnose between local recurrence and metastatic disease with calcitonin alone. It is also known that calcitonin levels reach undetectable levels slowly after the initial surgery. Therefore, morphological imaging examinations via Ultrasound and computed tomography (U/S and CT) play an important adjunctive role in the screening of these patients [4,11]. Ultrasonography, along with clinical examination, has been included in the necessary postoperative follow-up of these patients due to its high sensitivity for detecting local recurrence, while when calcitonin values exceed 150 pg/mL, CT is necessary to identify the presence of distant metastatic foci [11,12]. However, a problem remains in cases where calcitonin levels remain detectable but no morphological imaging disturbance can be observed. This condition is called biochemical incomplete response [13]. At this point, functional imaging procedures play an important role in differential diagnosis since this tumor is easily imaged with the administration of many radiopharmaceuticals, such as Fluoro-dihydroxyphenylalanine (F-DOPA), 18-FDG, and Ga68-DOTATATE [1,7,11]. Despite the significant help that PET/CT scanning provides in the detection of local recurrence or metastatic disease, the American Thyroid Association (ATA), in a revision of its guidelines in 2015, did not include PET in the necessary tests for the detection of local recurrence or metastases, even if calcitonin remains at detectable levels and no morphological imaging disturbance is observed, because it is thought that PET scanning is not particularly sensitive in the detection of the above cases [7]. In contrast, the 2019 guidelines from the European Society for Medical Oncology (ESMO) recommend PET/CT scanning in the category of biochemical incomplete response and even emphasize that, among other things, F-DOPA has high sensitivity and specificity in the detection of medullary thyroid carcinoma [14]. The only problems are the cost and availability of the test. This disagreement reached the point that the European Association for Nuclear Medicine (EANM) refused to give its consent for the revision of the ATA guidelines in 2015 [15]. This systematic review and meta-analysis aimed to fill the above gap. The sensitivity and specificity of the predominant PET radiopharmaceuticals in detecting local recurrence and metastases in surgical patients with MTC were comparatively studied. The objective was to provide new information on the above issue, especially from 2019 onwards, after the last revision of the European Society for Medical Oncology (ESMO) guidelines.

## 2. Materials and Methods

The study design was based on the “PICO” algorithm, as follows: patients: surgical patients with medullary thyroid carcinoma; intervention: PET scan for the detection of recurrence and metastases; comparison: F-DOPA, 18-FDG, and GA68-DOTATATE both with each other and with Ct and CT scanning; outcome: high sensitivity and specificity, especially for-DOPA in the demonstration of recurrence and metastases in surgical MTC patients in biochemical incomplete response after operations on thyroid carcinoma. Based on this design, this research was conducted through a database of three searchable sources: PubMed, Science Direct, and Research Gate. The combination of keywords used to search the literature was as follows: (A) “positron emission tomography (PET SCAN)”; (B) “computed tomography (CT)”; and (C) “medullary thyroid carcinoma”. Articles written in English were selected and cited in a reference list. Identical articles that appeared in the search sources and those that were not relevant to the study topic were excluded. Clinical trials, case reports, articles from encyclopedias, book chapters, conference abstracts, and mini critical reviews were also excluded. The analysis was performed based on the sensitivity and specificity of each radioactive material. This review was conducted in accordance with the Preferred Reporting Items for Systematic Reviews and Meta-Analyses (PRISMA) guidelines to ensure clarity and transparency. A detailed PRISMA flow diagram is included to illustrate the study selection process.

## 3. Results

A comprehensive online search was carried out after all articles were extensively cross-checked, resulting in 575 bibliographic studies up to the date of 20 June 2024. In total, 73 articles from PubMed, 47 articles from Research Gate, and 455 articles from Science Direct were included. After excluding duplicates (23 articles), irrelevant topics (174 articles), and all other criteria mentioned previously, such as not presenting the sensitivity and specificity of each radiopharmaceutical and not analyzing quantitative data, 7 studies [8,9,16,17,18,19,20] were finally assessed as suitable for our meta-analysis. The inclusion criteria included studies in which patients were classified as having biochemical incomplete response. This classification required a minimum of 6 months to have passed, during which calcitonin (Ct) levels remained detectable and no significant morphological abnormalities were observed in imaging tests. Only studies in which calcitonin (Ct) values were detectable over an extended period were included. The exclusion criteria included articles that did not report quantitative data and those that did not report on the sensitivity and specificity of each radiopharmaceutical [Figure 1 and Figure 2].

### Statistical Analysis

Five studies that recorded the diagnostic accuracy of specific diagnostic methods were included in the analysis, of which two were attributed separately to three of the five studies, leading to the recording of eight different estimates. The extracted data are presented in detail in Table 1.

An aggregated assessment of the measurements indicated that the prevalence was 0.79 [Table 2].

Table 3 shows an aggregated assessment of the sensitivity and specificity, as estimated for all studies included in the analysis. The results show that the sensitivity, regardless of methodology, was 82.8% with a 95% confidence interval of 75.3–88.3% for this assessment. In a similar way, the specificity, regardless of methodology, was 91.5% with a 95% confidence interval of 81.2–96.4% for this assessment. The Diagnostic Odds Ratio (DOR), regardless of methodology, was 51.91% with a 95% confidence interval of 18.66–144.37% for this assessment. It is noted that this confidence interval is particularly wide. The Positive Likelihood Ratio (LR+), determined as the probability of a positive result given a positive test, regardless of methodology, was 9.77% with a 95% confidence interval of 4.21–22.68% for this assessment. Note that values above 10 are considered too high for a clear conclusion. The Negative Likelihood Ratio (LR−), determined as the probability of a negative result given a negative test, regardless of methodology, was 0.19 with a 95% confidence interval of 0.13–0.28 for this assessment, and finally, the False Positive Rate (FPR), regardless of methodology, was 0.085% with a 95% confidence interval of 0.036–0.188% for this assessment, which shows that the probability of a false alarm was not negligible.

The heterogeneity between the studies for sensitivity was estimated to be 32.1%, and that of specificity was estimated to be 0%. The following [Fig cancers-16-04236-ch001] and [Fig cancers-16-04236-ch002] show the aggregated assessments of sensitivity and specificity, as estimated for all studies that were recorded. We note that in only two of them, namely Ertan Sahin 2020 and the Erdem Sahin 2020 subgroup study, sensitivity had values lower than 0.5, indicating that in these studies the findings were not statistically significant. For specificity, although ultimately the conclusion led to a significant differentiation from chance, it appears that the findings were statistically significant in only two of the individual studies before synthesis, and this shows the importance of synthesis.

Mapping the diagnostic accuracy through the pooled ROC curve showed that the confidence interval was extended in the upper left part, indicating good diagnostic value, although the specificity values marginally reached the value of 0.50 [[Fig cancers-16-04236-ch003]].

Based on the assessments of the Ga68-DOTATATE and 18-FDG methods presented in Table 4, the differences appeared to be small, as in all cases the confidence intervals of the two methods were overlapping. These indications were verified by a statistical test that followed for the sensitivity, specificity, and overall discriminatory power of the two methods. The statistical comparison values were *p* = 0.619, *p* = 0.868, and *p* = 0.859 for sensitivity, specificity, and overall diagnostic value, respectively [Table 5]. The non-differences are also graphically illustrated by the comparative ROC Summary of the two methods shown in [Fig cancers-16-04236-ch004]. Note that the F-DOPA method was excluded from the comparisons, as it had data from only two studies, and statistical comparison was not possible with less than three studies per category.

A meta-analysis of diagnostic accuracy was performed using the software Meta—DiSc, Version: 2.0 (Universidad Complutense, Barcelona, Spain), which led to the aggregate assessments and the design of the SROC. All estimates were based on the True Positive (TP), True Negative (TN), False Positive (FP), and False Negative (FN) data reported in the primary analyses of the studies. A random effects model established via the DerSimonian–Laird method was used for the analysis. The funnel plot [[Fig cancers-16-04236-ch005]] and Egger’s test were implemented using the meta, metabin, and funnel commands in R. The significance level was set to 0.05 in all cases [21,22].

Egger’s test (t = −2.18; *p* = 0.0719) for publication bias was non-significant, while the funnel plot was quite symmetrical [[Fig cancers-16-04236-ch005]], providing evidence in favor of the conclusion that the selection of the specific studies did not have a large effect on the results.

## 4. Discussion

Medullary thyroid carcinoma (MTC) is a rare neuroendocrine tumor that presents unique challenges in its diagnosis and management because of its often-aggressive nature and tendency to present with metastatic disease or local recurrence at the time of diagnosis [23]. In the last decade, the incorporation of advanced imaging techniques has revolutionized the approach to the diagnosis, staging, and monitoring of this type of carcinoma. Among them, positron emission tomography/computed tomography (PET-CT scanning) has emerged as a versatile tool that offers functional and anatomical insights into tumor behavior and the response to treatment.

It should be underlined that three conclusions can be drawn from the meta-analysis. Firstly, PET/CT scanning remains a highly sensitive and specific method for the detection of medullary thyroid carcinoma, regardless of the radiopharmaceutical used, with a sensitivity value of 82.8% (95% confidence interval: 75.3–88.3%) and a specificity value of 91.5% (95% confidence interval: 81.2–96.4%). In the meta-analysis that was performed, this was proven, and the corresponding ROC curve is presented, which emphasizes this conclusion. In addition, the “unappreciated” radiopharmaceuticals 18-FDG and Ga68-DOTATATE are also highly sensitive and reliable for the detection of MTC. The differences between the two methods (Ga-DOTATATE and 18-FDG) are small, and the confidence intervals overlap, indicating that there is no significant difference in the performance of the two methods. The sensitivity of the radiopharmaceutical Ga68-DOTATATE reached 83.2% (95% confidence interval: 70.1–91.3%), and that of the tracer 18-FDG reached 79.2% (95% confidence interval: 65.0–88.6%). Furthermore, with regards to specificity, Ga68-DOTATATE reached 92.7% (95% confidence interval: 60.4–99.1%), and 18-FDG reached 91.1% (95% confidence interval: 62.3–98.4%). The statistical comparison values (*p* = 0.619 for sensitivity, *p* = 0.868 for specificity, and *p* = 0.859 for the overall diagnostic value) indicate that the differences are not statistically significant and that the two methods are almost equally effective [8,9,16,17,18,19,20].

F-DOPA remains the most sensitive test and the most sensitive radiopharmaceutical for the detection of medullary thyroid carcinoma, which is also widely known. The sensitivity and specificity of these methods as a whole, and especially those of F-DOPA, provide comparable results to calcitonin alone and even have the advantage of accurate localization of metastases and local recurrence. Compared to U/S, Magnetic Resonance Imaging (MRI), and CT, it has the unique ability to precisely highlight metabolic activity and metastases, while the above-mentioned methods highlight only the morphological disorder [24]. This was confirmed by several studies, including both the meta-analysis by Lee at al. [25] and the systematic umbrella review by Trimboli et al. [26], who showed that among different imaging modalities, F-DOPA has the highest performance in detecting recurrent lesions in patients with MTC. Furthermore, in the comparative quantitative study by Asa et al. [9], in which 46 patients (25 women and 21 men) with medullary thyroid carcinoma were studied during postoperative follow-up, it was shown that the sensitivity and specificity rates of F-DOPA, on a per-patient basis, reached 86.8% and 100%, with a mean calcitonin value of 2031.9 pg/mL and a mean Carcinoembryonic Antigen (CEA) value of 68.3 ng/mL. Finally, in the most recent 15-year quantitative study in which 109 patients with MTC were studied, F-DOPA PET/CT was of great value for diagnosis and postoperative evaluation, with sensitivity and specificity rates of 95% and 93%, respectively, a mean calcitonin value of 1808 pg/mL, and a mean CEA value of 4 μg/L [27].

Despite all the studies mentioned above, development of therapeutic tracers that could improve the detection and management rates of MTC is still needed. The prevalence of stroma in most thyroid cancers presents new opportunities for targeting cancer-associated fibroblasts (CAFs) in molecular imaging and therapy. In recent years, this has included the Gallium-68 DOTA Small-Molecule Inhibitor of Fibroblast Activation Protein (68Ga-DOTA.SA.FAPi), which has shown an important therapeutic role related to the fibroblast protein activation inhibitor (FAPi) in various cancers, including medullary thyroid carcinoma [28]. This was also highlighted in a comparative quantitative study by Ballal et al., in which 27 patients with MTC were evaluated. The radiopharmaceuticals 68Ga-DOTA.SA.FAPi and Gallium-68 DOTANOC (Ga68-DOTANOC) showed comparable detection rates for the diagnosis of primary tumors (100% [18 of 18] vs. 94.4% [17 of 18], *p* = 0.979), lymph node involvement (98.3% [118 of 120] vs. 95% [114 of 120], *p* = 0.288), and brain metastases (100%). However, [68Ga] (Ga-DOTA.SA.FAPi) demonstrated significantly higher sensitivity rates compared to [68Ga] Ga-DOTANOC PET/CT for the detection of lung nodules (93.5% [87 of 93] vs. 68.9% [64 of 93]), liver metastases (100% [105 of 105] vs. 46.4% [49 of 105]), bone metastases (92.4% [110 of 119] vs. 76.5% [91 of 111] *p* = 0.001), and pleural metastases (98.2% vs. 0%) [29].

One radiopharmaceutical worth noting is sodium fluoride (18F-NaF), which has caused much controversy in the past regarding its possible inclusion in guidelines. Thus, Treglia and Luca Giovanella et al. reported that its use is far from proven in terms of the investigation and localization of skeletal metastatic foci, which justifies its exclusion from the current EANM guidelines for PET/CT scanning [30]. Since then, however, a comparative quantitative study by Ueda et al., which analyzed 31 patients with NSCLC, reported that the use of the tracer 18F-NaF is equal or superior to other imaging modalities such as bone scintigraphy, 18-FDG- PET/CT, Ga68-DOTATATE PET/CT, CT, and MRI in the detection of metastatic bone lesions and allows analysis of the whole skeleton in a single study [31]. Finally, the disadvantages include that PET/CT scanning is not cost-effective because it is a very expensive test and that it is not available at all testing centers [7,32].

However, our research has some limitations. This is evidenced by the fact that there were few quantitative surveys in this 5-year period (2020–2024). First of all, for F-DOPA, which is the most widely studied radiopharmaceutical, we had few studies available during this period of time. Specifically, we had two studies. This led us to look at all the studies we had together to come up with average serum sensitivity and specificity values for PET/CT scanning overall independent of the radiopharmaceutical used. However, a subgroup analysis was performed for18-FDG and Ga68-DOTATATE, and it was found that they did not differ significantly in terms of sensitivity and specificity. This comparison between these two radiopharmaceuticals was carried out because more studies were available. Nevertheless, F-DOPA is known from earlier studies to be the most effective radiopharmaceutical.

Another limitation is related to the inclusion of CEA as a biomarker. While CEA is a highly reliable marker, its inconsistent use across the included studies (with some studies incorporating it and others not) made it difficult to include it in the inclusion criteria. Incorporating CEA would have undermined the reliability of our analysis due to the insufficient number of studies consistently reporting on it. This represents a limitation of the current review. We hope that a new systematic review and meta-analysis that includes CEA will be conducted [33].

## 5. Conclusions

This systematic review and meta-analysis provide a higher quality of evidence and a higher grade of recommendation for the existing recommendations of both the ESMO and the ATA. We hope that this meta-analysis will close the gap that exists between the two major organizations that draft guidelines and make it clear that the problem with PET/CT scanning is not its sensitivity, since it is presumed to be highly sensitive and specific, but some other obstacles, such as availability and cost-effectiveness, as we have already mentioned.

## Data Availability

Not applicable.

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
