# Peer review of "The Efficacy of Positron Emission Tomography/Computed Tomography Scan (PET CT Scan) in the Diagnosis of Local Recurrence and Metastases in Surgical Patients with Medullary Thyroid Carcinoma: A Systematic Review and Meta-Analysis of the Last 5 Years (2020–2024)"

_cancers, 2024, doi:10.3390/cancers16244236_

Round 1
Reviewer 1 Report
Comments and Suggestions for Authors
This is a systematic review about the possibility to use PET-CT scan to detect metastases of medullary thyroid carcinoma.
Only seven studies have been used for this purpose.
I agree with the Authors that F-DOPA is the gold standard rather than DOTANOC or FDG, even if FDG-PET is cheapear then the others.
I strongly agree that all patients with MTC should have the possibility to perform F-DOPA; I am not confident with FDG in this particular case, beacuse FDG is uptaken almost by everything, and not only from eventual MTC metastases.
I think that the statistical part is really complicated and should be ameliorated.
Author Response
Dear author,
Thank you for your thoughtful and constructive review of our manuscript. We appreciate your feedback and have carefully addressed each of the points you raised below.
This systematic review and meta-analysis focused on studies conducted over the past five years. This decision was made because:
- a) Previous guidelines and systematic reviews provided sufficient answers to this issue.
- b) Professor Iakovou, a member of the EANM, who contributed to drafting the guidelines and he was one of the main authors, studied them carefully and deemed it appropriate for this study to focus on the past five years.
It is well-established in the literature that F-DOPA is the most effective radiopharmaceutical. However, only two studies in the past five years have examined F-DOPA. These studies confirm its superiority as the best radiopharmaceutical, which is why it was not included in the meta-analysis.
Nonetheless, as we emphasized, other radiopharmaceuticals, such as FDG and GA-DOTATATE, are often underestimated for detecting medullary carcinoma. We placed strong emphasis on highlighting this fact.
Given that only three studies were available for this analysis, and because these radiopharmaceuticals are often misunderstood, we performed a comparative analysis. The conclusion drawn is that they are equally sensitive but less specific, as you also noted.
Regarding the statistical analysis, we employed the most advanced statistical methods with the support of an experienced statistical team. We thoroughly examined the heterogeneity of the studies as well as any potential biases. We do not believe that the extracted data significantly affect the results to the extent of misleading the scientific community. Regarding the statistical analysis methodology used in our meta-analysis. For the analysis of diagnostic accuracy, we performed a meta-analysis using Meta-DiSc software (Version: 2.0, Universidad Complutense, Barcelona, Spain). This software allowed us to calculate aggregate estimates and generate the summary receiver operating characteristic (SROC) curve.
The data for the analysis were derived from the true positives (TP), true negatives (TN), false positives (FP), and false negatives (FN) reported in the primary analyses of the included studies. We used a random effects model under the DerSimonian-Laird method to account for the variability between the studies.
Additionally, to assess publication bias, we employed Egger’s test and visualized the data with a funnel plot. These analyses were conducted using the meta, metabin, and funnel commands in R (a statistical computing environment).
We hope this clarifies the statistical methods used in our analysis. If you have any further questions or require additional details, please do not hesitate to ask.
With regard to the cost of the radiopharmaceuticals, this aspect is also described in great detail, particularly in the context of the overall PET/CT scan procedure.
We sincerely appreciate your time and valuable feedback.
Please let me know if there are any further adjustments or clarifications needed. I greatly appreciate your guidance throughout this process.
Best regards,
Dr. Konstantinos Papadopoulos
Reviewer 2 Report
Comments and Suggestions for Authors
Konstantinos Papadopoulos and co-authors have addressed the role of PET/CT in monitoring disease recurrence in medullary carcinoma in this meta-analysis that included seven eligible studies. The paper is clearly written, it brings together more recent studies, the study design was done according to the "PICO" algorithm, the discussion and conclusions are clear, as are the study limitations. My comments are as follows:
1. Did you notice during data extraction in which period calcitonin analysis was performed? Every how many months? At what interval from surgery?
2. CEA is also a monitoring marker, why did you not analyze it?
3. Figure 2 is not suitable for presentation in the main text, maybe in supplementary data?
4. The study limitations should be explained more clearly.
Otherwise, the current topic is interesting, congratulations to the authors.
Author Response
Dear Reviewer,
Thank you for your thoughtful comments and suggestions on our manuscript. We greatly appreciate your feedback. Below, we have provided our responses to the points you raised:
Biochemical Incomplete Response: All patients included in the study were categorized as having Biochemical Incomplete Response, meaning at least six months had passed during which calcitonin levels remained detectable, and no significant morphological abnormalities were observed in the imaging tests. We will include this in the inclusion criteria to clearly define the study group.
CEA as a Biomarker: While CEA is a reliable marker, its inclusion in the inclusion criteria was not feasible due to its inconsistent use in the studies we analyzed. Including CEA would have reduced the reliability of the research because of the insufficient number of studies reporting it consistently. We are considering a future systematic review and meta-analysis that will include CEA, and we will note this limitation in the current study.
Other Limitations: While there are additional limitations to consider, we have focused on the most significant ones.
Thank you once again for your valuable feedback. We believe the revisions will strengthen the manuscript, and we appreciate your thoughtful contributions to improving our work.
Kind regards,
Dr. Konstantinos Papadopoulos
Reviewer 3 Report
Comments and Suggestions for Authors
This is a systematic review and metaanalysis of studies using PET/CT for the diagnosis of medullary thyroid carcinoma recurrence or metastasis.
It aims to offer additional evidence to harmonise the discordant recommendations of 2 different organizations (ESMO and ATA) regarding the importance of PET/CT scan in the postoperative follow-up of MTC patients.
Minor issues:
1. Line 143, 166 the message[Error! Reference source not found.] appears
2.Figure 2 should be reviewed for readability
3.please provide an explanation for all abbreviations before using them first time, even for frequently used ones (e.g LR, DOR)
Strengths: the methods are clearly described and they demonstrate that PET/CT scan is a highly sensitive and specific method in the detection of the MTC recurrence or metastasis, regardless of the radiopharmaceutical used.
Major issue: why are the results for the 2 radiopharmaceuticals discussed not compared to those for the F DOPA since 2 studies using it have been included in the analysis?
Author Response
Dear Reviewer,
Thank you for your valuable feedback on our manuscript. We appreciate your thoughtful comments and suggestions, and we will address each point raised below:
Line 143, 166 – Error Message [Error! Reference source not found.]
We apologize for the formatting issue. We will correct these references and ensure that all citations are properly linked to their respective sources in the revised manuscript.
Figure 2 – Readability
Thank you for pointing this out. We will review Figure 2 and adjust the design to enhance its clarity and readability, ensuring that it is easily interpretable for readers.
Abbreviations
We understand the importance of clarity for all readers. In the revised version, we will ensure that all abbreviations, including LR (Likelihood Ratio) and DOR (Diagnostic Odds Ratio), are defined upon first use, even if they are commonly known, to maintain consistency and clarity throughout the manuscript.
Comparison of the Two Radiopharmaceuticals (FDG, GA-DOTATATE) with F-DOPA
Since the systematic review and meta-analysis was conducted over the last five years, and this was done following the guidance of Professor Iakovou, who co-drafted the guidelines with the EANM, the number of studies was limited. Specifically, for F-DOPA, there were only two studies (Araz, Asa). For this reason, due to the small sample size for F-DOPA, we did not compare it with the other radiopharmaceuticals. However, F-DOPA is a well-established radiopharmaceutical for the successful detection of MTC. Our focus was on the often misunderstood radiopharmaceuticals, FDG and DOTATATE, which require further research. Some conclusions can still be drawn for F-DOPA, given that its sensitivity and specificity are clearly demonstrated and presented in the forest plot (Araz, Asa). We hope to conduct a new meta-analysis that will include a larger number of studies, both for F-DOPA and the other radiopharmaceuticals, and this is described in detail in the limitations section of the manuscript.
Thank you again for your valuable comments and suggestions. We will carefully address each of the points you raised to improve the quality of our manuscript. We appreciate your time and effort in reviewing our work.
Kind regards,
Dr. Konstantinos Papadopoulos
Round 2
Reviewer 3 Report
Comments and Suggestions for Authors
This systematic review and metaanalysis of studies using PET/CT for the diagnosis of medullary thyroid carcinoma recurrence or metastasis has been improved compared to the initial version.
The message[Error! Reference source not found.] still appears!! (line 183)
2.Figure 2 is, in my opinion, of an inadequate quality for a journal but I leave this to the editorial office to decide
3. the fact that the analysis only included the last 5 years is only to be found in the title of the manuscript; why did the authors decide to limit the analysis as such?
Author Response
Dear Reviewer,
Thank you for your constructive feedback on our manuscript. We appreciate the opportunity to address your comments and make the necessary revisions to improve the quality and clarity of our work. Below, we provide detailed responses to each of the points raised.
1) We have carefully reviewed the manuscript, particularly around line 183, and found no explicit reference or citation that could cause the '[Error! Reference source not found.]' message. However, we suspect this might be due to a broken cross-reference for a figure or table. To resolve this, we have thoroughly updated all cross-references in the document. If the issue persists on your end, please let us know, and we will work with you to resolve it promptly. Let me know if you'd like further help troubleshooting this!
2) We acknowledge your concern about its quality. While we believe the figure effectively conveys the intended information, we understand the importance of meeting the journal's visual and formatting standards.
If the editorial office deems it necessary, we are prepared to revise or replace Figure 2 to enhance its quality or clarity as per the journal's guidelines. Please do let us know if any specific adjustments are required.
3) The decision to conduct a systematic review of the literature over the past five years was made by Professor Iakovou with the agreement of the research team. This decision was based on the following reasons:
- The effectiveness of FDOPA in diagnosing medullary thyroid carcinoma is already well-established.
- The relevant guidelines from the EANM, which were incorporated into the ESMO recommendations during the most recent revision, have been developed.
It should be noted that Professor Iakovou and his team are key members of the EANM and contributed significantly to drafting the aforementioned guidelines.
The aim was to examine the latest data and address the misconceptions surrounding PET, even for the "misunderstood" radiopharmaceuticals. This has been clearly described in the manuscript, both in the study criteria and in the Limitations section.
The PET/CT scan is a highly effective method for detecting local recurrence and distant metastases, particularly in patients with a biochemical incomplete response after surgery. We firmly believe that the study results have been accurately presented, and any minor limitations do not mislead the scientific community.
If the editorial team wishes, we are at your disposal to include all the above points in the manuscript, although we believe that the topic has already been described with thoroughness and clarity.
Thank you for your valuable feedback and guidance. We have addressed the concerns raised and remain available for any further clarifications or revisions required. We appreciate your consideration of our manuscript.
Kind regards,
Dr. Konstantinos Papadopoulos